# DISTRIBUTION REGRESSION NETWORK

## ABSTRACT

We introduce our Distribution Regression Network (DRN) which performs regression from input probability distributions to output probability distributions. Compared to existing methods, DRN learns with fewer model parameters and easily extends to multiple input and multiple output distributions. On synthetic and real-world datasets, DRN performs similarly or better than the state-of-the-art. Furthermore, DRN generalizes the conventional multilayer perceptron (MLP). In the framework of MLP, each node encodes a real number, whereas in DRN, each node encodes a probability distribution.

## 1 INTRODUCTION

The field of regression analysis is largely established with methods ranging from linear least squares to multilayer perceptrons. However, the scope of the regression is mostly limited to real valued inputs and outputs (Fiori et al., 2015; Marquardt, 1963). In this paper, we perform distribution-to-distribution regression where one regresses from input probability distributions to output probability distributions.

Distribution-to-distribution regression (see work by Oliva et al. (2013)) has not been as widely studied compared to the related task of functional regression (Ferraty & Vieu, 2006). Nevertheless, regression on distributions has many relevant applications. In the study of human populations, probability distributions capture the collective characteristics of the people. Potential applications include predicting voting outcomes of demographic groups (Flaxman et al., 2016) and predicting economic growth from income distribution (Perotti, 1996). In particular, distribution-to-distribution regression is very useful in predicting future outcomes of phenomena driven by stochastic processes. For instance, the Ornstein-Uhlenbeck process, which exhibits a mean-reverting random walk, has wide-ranging applications. In the commodity market, prices exhibit mean-reverting patterns due to market forces (Schwartz & Smith, 2000). It is also used in quantitative biology to model phenotypic traits evolution (Bartoszek et al., 2016).

Variants of the distribution regression task have been explored in literature (Póczos et al., 2013; Oliva et al., 2014). For the distribution-to-distribution regression task, Oliva et al. (2013) proposed an instance-based learning method where a linear smoother estimator (LSE) is applied across the input-output distributions. However, the computation time of LSE scales badly with the size of the dataset. To that end, Oliva et al. (2015) developed the Triple-Basis Estimator (3BE) where the prediction time is independent of the number of data by using basis representations of distributions and Random Kitchen Sink basis functions. Lampert (2015) proposed the Extrapolating the Distribution Dynamics (EDD) method which predicts the future state of a time-varying probability distribution given a sequence of samples from previous time steps. However, it is unclear how it can be used for the general case of regressing distributions of different objects.

Our proposed Distribution Regression Network (DRN) is based on a completely different scheme of network learning, motivated by spin models in statistical physics and similar to artificial neural networks. In many variants of the artificial neural network, the network encodes real values in the nodes (Rumelhart et al., 1985; LeCun et al., 1989; Bengio, 2009). DRN is novel in that it generalizes the conventional multilayer perceptron (MLP) by encoding a probability distribution in each node. Each distribution in DRN is treated as a single object which is then processed by the connecting weights. Hence, the propagation behavior in DRN is much richer, enabling DRN to represent distribution regression mappings with fewer parameters than MLP. We experimentally demonstrate that compared to existing methods, DRN achieves comparable or better regression performance with fewer model parameters.

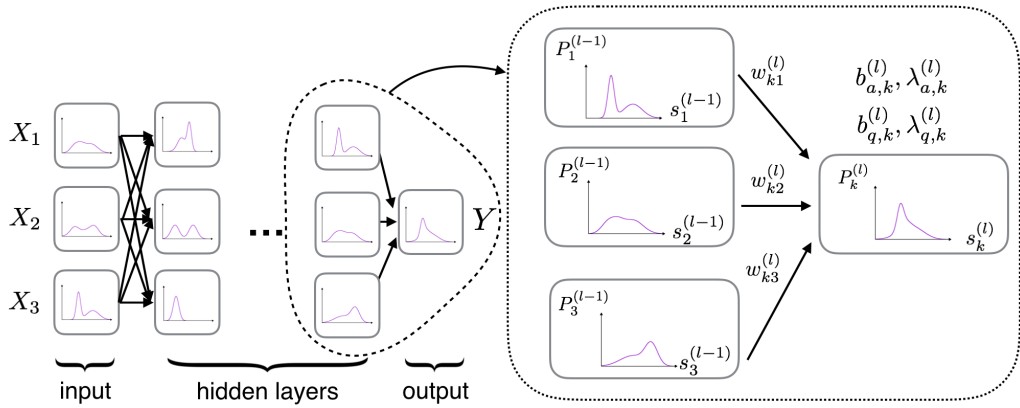

Figure 1: (Left) An example DRN with multiple input probability distributions and multiple hidden layers mapping to an output probability distribution. (Right) A connection unit in the network, with 3 input nodes in layer $l-1$ connecting to a node in layer $l$. Each node encodes a probability distribution, as illustrated by the probability density function $P_k^{(l)}$. The tunable parameters are the connecting weights and the bias parameters at the output node.

## 2 DISTRIBUTION REGRESSION NETWORK

Given a training dataset with $M$ data points $\mathcal{D} = \{(X_1^1, \cdots, X_1^K, Y_1), \cdots, (X_M^1, \cdots, X_M^K, Y_M)\}$ where $X_i^k$ and $Y_i$ are univariate continuous distributions with compact support, the regression task is to learn the function $f$ which maps the input distributions to the output distribution.

$$Y_i = f(X_i^1, \cdots, X_i^K) \tag{1}$$

No further assumptions are made on the form of the distribution. It is trivial to generalize our method to regress to multiple output distributions but for simplicity of explanation we shall restrict to single output regressions in the following discussions.

### 2.1 FORWARD PROPAGATION

Fig. 1 illustrates how the regression in Eq. (1) is realized. DRN generalizes the traditional neural network structure by encoding each node with a probability distribution and connecting the nodes with real-valued weights. The input data consists of one or more probability distributions which are fed into the first layer and propagated layerwise through the hidden layers. We emphasize our network is not a Bayesian network even though each node encodes a probability. Unlike bayes net where the conditional probability among variables are learnt by maximizing the likelihood over observed data, DRN regresses probability distributions using a feedforward network, similar to MLP.

At each node in the hidden layer, the probability distribution is computed from the probability distributions of the incoming nodes in the previous layer and the network parameters consisting of the weights and bias parameters (see right of Fig. 1). $P_k^{(l)}$ represents the probability density function (pdf) of the $k^{\text{th}}$ node in the $l^{\text{th}}$ layer and $P_k^{(l)}(s_k^{(l)})$ is the density of the pdf when the node variable is $s_k^{(l)}$.

Before obtaining the probability distribution $P_k^{(l)}$, we first compute its unnormalized form $\tilde{P}_k^{(l)}$. $\tilde{P}_k^{(l)}$ is computed by marginalizing over the product of the unnormalized conditional probability $\tilde{Q}(s_k^{(l)}|s_1^{(l-1)}, \cdots, s_n^{(l-1)})$ and the incoming node probabilities.

$$\tilde{P}_k^{(l)}\left(s_k^{(l)}\right) = \int_{s_1^{(l-1)}} \cdots \int_{s_n^{(l-1)}} \tilde{Q}\left(s_k^{(l)}|s_1^{(l-1)}, \cdots, s_n^{(l-1)}\right) \tag{2}$$

$$P_1^{(l-1)}\left(s_1^{(l-1)}\right) \cdots P_n^{(l-1)}\left(s_n^{(l-1)}\right) ds_1^{(l-1)} \cdots ds_n^{(l-1)}$$

$$\tilde{Q}\left(s_k^{(l)}|s_1^{(l-1)}, \cdots, s_n^{(l-1)}\right) = \exp\left[-E\left(s_k^{(l)}|s_1^{(l-1)}, \cdots, s_n^{(l-1)}\right)\right] \tag{3}$$

$s_1^{(l-1)}, \cdots, s_n^{(l-1)}$ represent the variables of the lower layer nodes and $E$ is the energy given a set of node variables, which we define later in Eq. (4). The unnormalized conditional probability has the same form as the Boltzmann distribution in statistical mechanics, except that the partition function is omitted. This omission reduces the computational complexity of our model through factorization, shown later in Eq. (5).

Our energy function formulation is motivated by work on spin models in statistical physics where spin alignment to coupling fields and critical phenomena are studied (Lee et al., 2002; 2003; Katsura, 1962; Wu, 1982). Energy functions are also used in other network models where a scalar energy is associated to each configuration of the nodes (Teh et al., 2003; LeCun et al., 2007). In such energy-based models, the parameters are learnt such that the observed configurations of the variables have lower energies than unobserved ones. However, the energy function used in DRN is part of the forward propagation process and is not directly optimized. For a given set of node variables, the energy function is

$$E\left(s_k^{(l)}|s_1^{(l-1)}, \cdots, s_n^{(l-1)}\right) = \sum_i^n w_{ki}^{(l)}\left(\frac{s_k^{(l)} - s_i^{(l-1)}}{\Delta}\right)^2 + b_{q,k}^{(l)}\left(\frac{s_k^{(l)} - \lambda_{q,k}^{(l)}}{\Delta}\right)^2$$
$$+ b_{a,k}^{(l)}\left|\frac{s_k^{(l)} - \lambda_{a,k}^{(l)}}{\Delta}\right| \tag{4}$$

$w_{ki}^{(l)}$ is the weight connecting the $i^{\text{th}}$ node in the lower layer to the upper layer node. $b_{q,k}^{(l)}$ and $b_{a,k}^{(l)}$ are the values of the quadratic and absolute bias terms which act at the positions $\lambda_{q,k}^{(l)}$ and $\lambda_{a,k}^{(l)}$ respectively. The support length of the distribution is given by $\Delta$. All terms in Eq. (4) are normalized by the support length so that the energy function is invariant with respect to the support. Eq. (2) can be factorized such that instead of having multidimensional integrals, there are $n$ univariate integrals:

$$\tilde{P}_k^{(l)}\left(s_k^{(l)}\right) = \exp\left(B\left(s_k^{(l)}\right)\right)\int_{s_1^{(l-1)}} \cdots \int_{s_n^{(l-1)}} P_1^{(l-1)}\left(s_1^{(l-1)}\right) \cdots P_n^{(l-1)}\left(s_n^{(l-1)}\right) \tag{5}$$
$$\exp\left[-\sum_i^n w_{ki}^{(l)}\left(\frac{s_k^{(l)} - s_i^{(l-1)}}{\Delta}\right)^2\right] ds_1^{(l-1)} \cdots ds_n^{(l-1)}$$
$$= \exp\left(B\left(s_k^{(l)}\right)\right)\prod_i^n\left\{\int_{s_i^{(l-1)}} P_i^{(l-1)}\left(s_i^{(l-1)}\right)\exp\left[-w_{ki}^{(l)}\left(\frac{s_k^{(l)} - s_i^{(l-1)}}{\Delta}\right)^2\right] ds_i^{(l-1)}\right\}$$

where $B(s_k^{(l)})$ captures the bias terms of the energy function in Eq. (4).

$$B\left(s_k^{(l)}\right) = -b_{q,k}^{(l)}\left(\frac{s_k^{(l)} - \lambda_{q,k}^{(l)}}{\Delta}\right)^2 - b_{a,k}^{(l)}\left|\frac{s_k^{(l)} - \lambda_{a,k}^{(l)}}{\Delta}\right| \tag{6}$$

Finally, the probability distribution from Eq. (2) is normalized.

$$P_k^{(l)}\left(s_k^{(l)}\right) = \frac{\tilde{P}_k^{(l)}\left(s_k^{(l)}\right)}{\int_{s_k^{(l)'}} \tilde{P}_k^{(l)}\left(s_k^{(l)'}\right) ds_k^{(l)'}} \tag{7}$$

The propagation of probability distributions within a connection unit forms the basis for forward propagation. Forward propagation is performed layerwise from the input layer using Eq. (2) to (7).

### 2.1.1 PROPAGATION PROPERTIES

The forward propagation in DRN has some important properties. Fig. 2 illustrates the propagation behavior for a connection unit with one input node where the bias values $b_{a,k}^{(l)}$ and $b_{q,k}^{(l)}$ are set as zero.

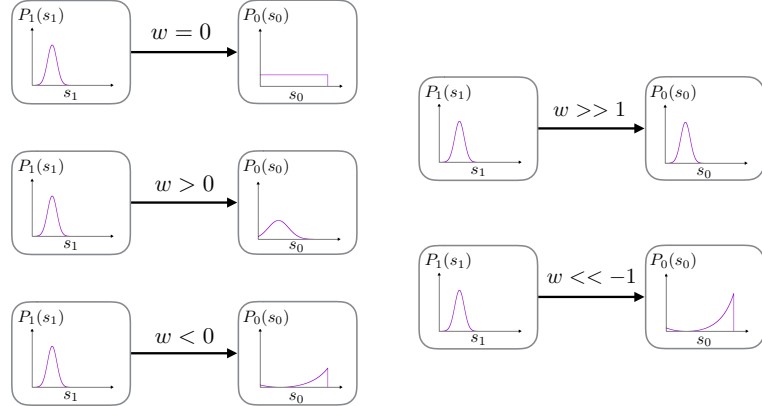

Figure 2: Propagation behavior for a connection unit with one input node. The biases are set as zero in these examples. When weight is zero, the output distribution is flat. Positive weights causes the output distribution to have the same peak position as the input distribution while negative weights causes the output pdf to 'repel' away from the input peak. When the weight is a sufficiently large positive number, the propagation tends towards the identity mapping.

When the weight is zero, the output distribution is flat and the output distribution is independent of the input. With a positive weight, the output distribution is 'attracted' to the peak of the input distribution whereas a negative weight causes the output distribution to be 'repelled' away from the input peak. In addition, the weight magnitude represents the strength of the 'attraction' or 'repulsion'. When the weight is a sufficiently large positive number, the propagation tends towards the identity mapping (top right example in Fig. 2). The implication is that like in neural networks, a deeper network should have at least the same complexity as a shallow one, as the added layers can produce the identity function. Conversely, a small positive weight causes the output peak to be at the same position as the input peak, but with more spread (second example on left column of Fig. 2).

The remaining absolute and quadratic bias terms in Eq. (4) have a similar role as the bias in a traditional neural network. Depending on the bias values $b_{a,k}^{(l)}$ and $b_{q,k}^{(l)}$, the bias terms act as attractors or repellers from the positions defined by $\lambda_{a,k}^{(l)}$ and $\lambda_{q,k}^{(l)}$ respectively. The weight and bias values play a similar role as the inverse temperature in the Boltzmann distribution in statistical physics (Lee et al., 2002; Wu, 1982).

## 2.2 Network cost function

The cost function of the network given a set network parameters is measured by the Jensen-Shannon (JS) divergence between the label ($Y_i$) and predicted ($\hat{Y}_i$) distributions. The JS divergence is given by $D_{JS}(Y_i||\hat{Y}_i) = \frac{1}{2}D_{KL}(Y_i||W_i) + \frac{1}{2}D_{KL}(\hat{Y}_i||W_i)$, where $W_i = \frac{1}{2}(Y_i + \hat{Y}_i)$ and $D_{KL}$ is the Kullback-Liebler divergence. The Jensen-Shannon divergence is a suitable cost function as it is symmetric and bounded. The network cost function $C_{net}$ is the average $D_{JS}$ over all $M$ training data: $C_{net} = \frac{1}{M}\sum_i^M D_{JS}(Y_i||\hat{Y}_i)$.

## 2.3 Discretization of Probability Distributions

In our experiments, the integrals in Eq. (5) and (7) are performed numerically. This is done through discretization from continuous probability density functions (pdf) to discrete probability mass functions (pmf). Given a continuous pdf with finite support, the range of the continuous variable is partitioned into $q$ equal widths and the probability distribution is binned into the $q$ states. The estimation error arising from the discretization step will decrease with larger $q$.

## 2.4 Optimization by Backpropagation

The network cost is a differentiable function over the network parameters. We derive the cost gradients similar to backpropagation in neural networks (Rumelhart et al., 1988). We use chain rule to derive at each node a $q$-by-$q$ matrix which denotes the derivative of the final layer node distribution with respect to the current node distribution.

$$\frac{\partial P_1^{(L)}\left(s_1^{(L)}\right)}{\partial P_k^{(l)}\left(s_k^{(l)}\right)} = \sum_i^n \sum_{s_i^{(l+1)}} \frac{\partial P_1^{(L)}\left(s_1^{(L)}\right)}{\partial P_i^{(l+1)}\left(s_i^{(l+1)}\right)} \frac{\partial P_i^{(l+1)}\left(s_i^{(l+1)}\right)}{\partial P_k^{(l)}\left(s_k^{(l)}\right)} \tag{8}$$

where $P_1^{(L)}\left(s_1^{(L)}\right)$ is the final layer output probability distribution. From the derivative $\frac{\partial P_1^{(L)}\left(s_1^{(L)}\right)}{\partial P_k^{(l)}\left(s_k^{(l)}\right)}$, the cost gradients for all network parameters can be obtained. Detailed derivations of the cost gradients are included in Appendix A. The network weights $w_{ki}^{(l)}$ and bias magnitudes $b_{a,k}^{(l)}$, $b_{q,k}^{(l)}$ are randomly initialized with a uniform distribution, though other initialization methods are also feasible. The bias positions $\lambda_{a,k}^{(l)}$ and $\lambda_{q,k}^{(l)}$ are uniformly sampled from the range corresponding to the support of the distributions.

## 3 Experiments

We evaluate DRN on synthetic and real-world datasets and compare its performance to the state-of-the-art 3BE method and a fully-connected multilayer perceptron (MLP). For each of the datasets, DRN achieves similar or higher accuracy with fewer model parameters.

In MLP, each discretized probability mass function is represented by $q$ nodes. The MLP consists of fully connected hidden layers with ReLU units and a softmax final layer, and is optimized with mean squared error using Adam. Unlike DRN and MLP where the distribution pdfs are directly used by the methods, 3BE assumes the input and output distributions are observed through i.i.d. samples. Hence, for the first two datasets we provide 3BE with sufficient samples from the underlying distribution such that errors from density estimation are minimal.

## 3.1 Synthetic Data

The first experiment involves a synthetic dataset similar to the one used by Oliva et al. (2013) but with increased complexity. We first generate two truncated gaussians by sampling their means $\mu_1 \sim \text{Unif}[0.1, 0.4]$, $\mu_2 \sim \text{Unif}[0.6, 0.9]$ and standard deviations $\sigma_1, \sigma_2 \sim \text{Unif}[0.05, 0.1]$. The input pdf is $X(s) = \gamma g(s; \mu_1, \sigma_1) + (1 - \gamma)g(s; \mu_2, \sigma_2)$ and the output pdf is $Y(s) = \gamma g(s; h(\mu_1, 0.1, 0.4), h(\sigma_1, 0.05, 0.1)) + (1 - \gamma)g(s; h(\mu_2, 0.6, 0.9), h(\sigma_2, 0.05, 0.1))$ where $g$ is the truncated normal pdf with support of [0,1], $\gamma \sim \text{Unif}[0, 1]$, and $h$ is

$$h(\nu, \nu_{min}, \nu_{max}) = \nu_{min} + \frac{\sin\left(2\pi \frac{\nu - \nu_{min}}{\nu_{max} - \nu_{min}}\right) + 1}{2} \times (\nu_{max} - \nu_{min}) \tag{9}$$

The function $h$ transforms the means and standard deviations using the non-linear function shown in Fig. 3a. The transformation is such that the two gaussian means will remain in their respective ranges. The sample input-output data pairs in Fig. 3b shows the complexity of the regression task with various behavior like peak splitting and peak spreading. 1000 training data and 1000 testing data were created to evaluate the regression methods.

For DRN and MLP, the pdfs are discretized into $q = 100$ states and for 3BE, 10,000 samples from each data distribution are generated. While 3BE gives a continuous distribution as the output, DRN and MLP output the discrete pmf and require conversion to continuous pdf. Following Oliva et al. (2014), the regression performance on the test set is measured by the $L2$ loss between the continuous predicted distribution, $\hat{Y}(s)$ and the true distribution.

We study how the regression accuracy varies with respect to the number of model parameters. For DRN and MLP, the number of parameters are varied using different depths and widths of the networks and for 3BE, we vary the number of Random Kitchen Sink features. We present the detailed

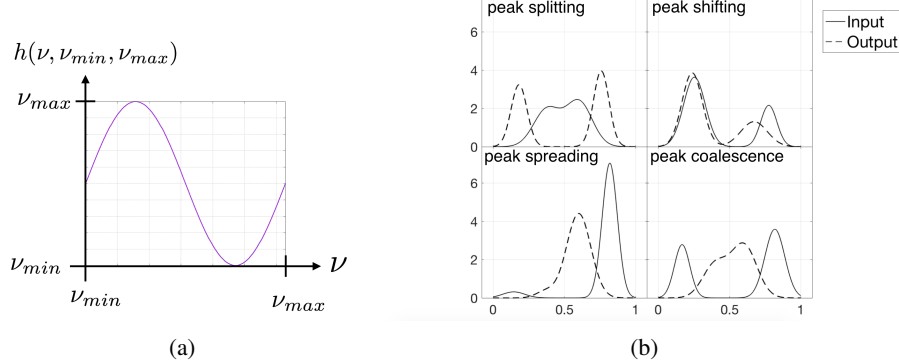

(a)                                                              (b)

Figure 3: (a) Nonlinear transformation of the input means and standard deviations of gaussians for the synthetic dataset. (b) Example input-output pairs from the synthetic data, illustrating the complexity of the regression task.

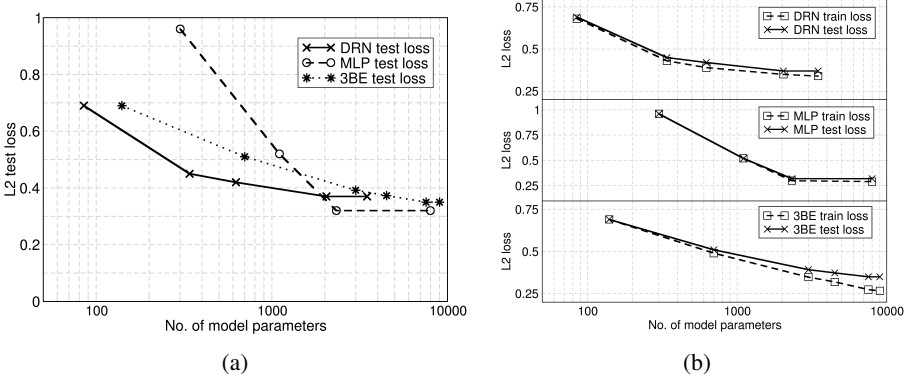

(a)                                                              (b)

Figure 4: (a) Comparison of $L2$ loss on the synthetic data test set. Note that the x-axis denotes the number of model parameters using the log scale. (b) Train and test loss for the individual methods as number of model parameters increases. There is no overfitting as the gaps between train and test losses are not significant.

DRN architecture in Appendix B. Fig. 4a shows the $L2$ loss on the test set as we vary the number of model parameters. Note that the x-axis is presented on the log scale. DRN's test performance is comparable to the other methods and uses fewer model parameters to attain reasonable performance. We note there is little overfitting for the three methods, as shown in the plots comparing train and test loss in Fig. 4b, though 3BE starts to exhibit overfitting when the number of model parameters approaches 10,000.

### 3.2 ORNSTEIN-UHLENBECK PROCESS

Because of the Boltzmann distribution term (ref. Eq. 3), DRN models the diffusion process very well. For this experiment, we evaluate our model on data generated from the stochastic Ornstein-Uhlenbeck (OU) process (Uhlenbeck & Ornstein, 1930) which combines the notion of random walk with a drift towards a long-term mean. The OU process has wide-ranging applications. In the commodity market, prices exhibit mean-reverting patterns due to market forces and hence modelling the prices with the OU process helps form valuation strategies (Schwartz & Smith, 2000; Zhang et al., 2012).

The OU process is described by a time-varying gaussian pdf. With the long-term mean set at zero, the pdf has a mean of $\mu(t) = y \exp(-\theta t)$ and variance of $\sigma^2(t) = \frac{D(1-e^{-2\theta t})}{\theta}$. $t$ represents time, $y$ is the initial point mass position, and $D$ and $\theta$ are the diffusion and drift coefficients respectively. The

regression task is to map from an initial gaussian distribution at $t_{init}$ to the resulting distribution after some time step $\Delta t$. The gaussian distributions are truncated with support of $[0, 1]$. With different sampled values for $y \in [0.3, 0.9]$ and $t_{init} \in [0.01, 2]$, pairs of distributions are created for $\Delta t = 1$, $D = 0.003$ and $\theta = 0.1$. For DRN and MLP, $q = 100$ was used for discretization of the pdfs while 10,000 samples were taken for each distribution to train 3BE.

Table 1: Comparison of $L2$ test loss and the number of model parameters used for the Ornstein-Uhlenbeck data.

|  | $L2$ test loss | Model description | No. of parameters |
|---|---|---|---|
| DRN | $0.1441 \pm 0.0010$ | No hidden layer | 5 |
| MLP | $0.1475 \pm 0.0005$ | 1 hidden layer of 3 nodes | 703 |
| 3BE | $0.1255 \pm 0.0083$ | 16 projection coefficients, 17 Random Kitchen Sink features | 272 |

We compare the number of model parameters required to achieve a small $L2$ test loss with 100 training data. We also increased the training size to 1000 and attained similar results. Table 1 and Fig. 5b show that a simple DRN of one input node connecting to one output node with 5 parameters performs similarly as MLP and 3BE. MLP requires 1 fully-connected hidden layer with 3 nodes, with a total of 703 network parameters. 3BE requires 64 projection coefficients for both input and output distributions and 17 Random Kitchen Sink features, resulting in 272 model parameters.

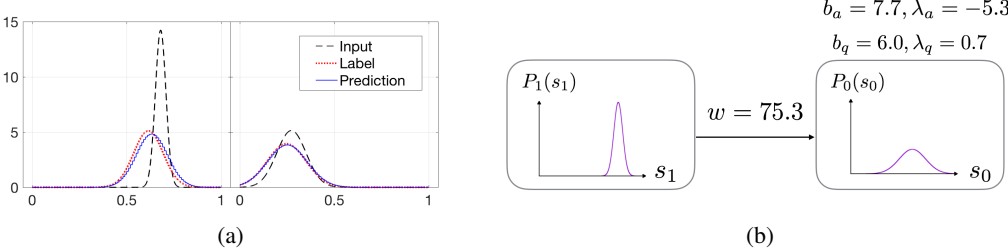

(a)  (b)

Figure 5: (a) The regression by DRN on two test samples. (b) The learnt parameters for DRN are interpreted as follows. The positive weight of 75.3 reflects the positive correlation between input and output peak positions and that the peak spreads out over time. The negative position of the absolute bias ($\lambda_a$) shows that the output peak is displaced leftwards of the input peak.

The regression by DRN on two random test samples are shown in Fig. 5a and we see that DRN is able to demonstrate the OU process. Fig. 5b shows the 5 DRN parameters after training. The values of these parameters are interpreted as follows. The weight parameter is positive, hence the output peak position is positively correlated to the input peak position. Moreover, $w = 75.3$ is such that the network mimics the diffusion property of the OU process. The bias position $\lambda_a$ is negative and its magnitude is 5 times the distribution support, causing the output peak to be displaced leftwards of the input peak. These two observations reflect the random walk and mean-reverting properties of the OU process.

Table 2: Comparison of log-likelihood on the stock data and the number of model parameters.

|  | Log-likelihood on test set | Model description | No. of parameters |
|---|---|---|---|
| DRN | $474.43 \pm 0.01$ | No hidden layer (Fig. 6) | 7 |
| MLP | $471.50 \pm 0.08$ | 1 hidden layer of 10 nodes | 4110 |
| 3BE | $466.76 \pm 0.73$ | 18 projection coefficients, 450 Random Kitchen Sink features | 8100 |

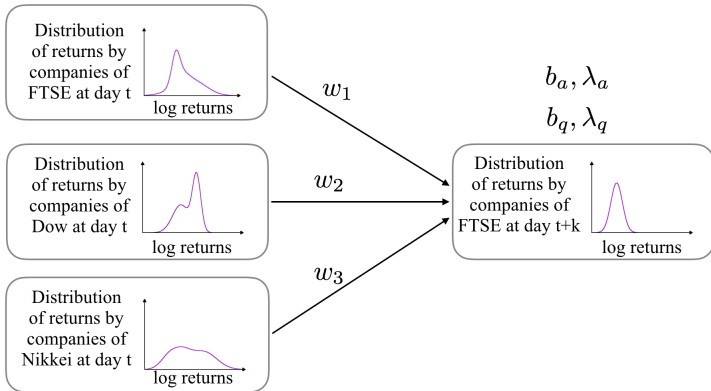

Figure 6: Single-layer network used in DRN for the stock dataset with 7 model parameters (3 weights, 4 bias parameters).

## 3.3 STOCK DATA

We demonstrate that DRN can be useful for an important real-world problem and outperforms 3BE and MLP in terms of prediction accuracy. With greater integration of the global stock markets, there is significant co-movement of stock indices (Hamao et al., 1990; Chong et al., 2008). In a study by Vega & Smolarski (2012), it was found that the previous day stock returns of the Nikkei and Dow Jones Industrial Average (Dow) are good predictors of the FTSE return. Modelling the co-movement of global stock indices has its value as it facilitates investment decisions.

Stock indices are weighted average of the constituent companies' prices in a stock exchange, and existing research has primarily focused on the movement of returns of the indices. However, for our experiment, we predict the future distribution of returns over the constituent companies in the index as it provides more information than just a weighted average. Our regression task is as follows. Given the current day's distribution of returns of constituent companies in FTSE, Dow and Nikkei, predict the distribution of returns for constituent companies in FTSE $k$ days later. The logarithmic return for the company's stock at day $t$ is given by $\ln(V_t/V_{t-1})$, where $V_t$ and $V_{t-1}$ represent its closing price at day $t$ and $t-1$ respectively.

The stock data consists of 9 years of daily returns from January 2007 to December 2015. To adapt to changing market conditions, we use a sliding-window training scheme where the data is split into windows of training, validation and test sets and moved foward in time (Kaastra & Boyd, 1996). A new window is created and the network is retrained after every 300 days (which is the size of test set). For each test set, the previous 500 and 100 days were used for training and validation. To reduce the noise in the data, we performed exponential window averaging on the price series for each stock with a window of 50 days following common practice (Murphy, 1999). The logarithmic returns of the constituent company stocks form the samples for the distributions of the returns.

For DRN and MLP, the pdf is estimated using kernel density estimation with a gaussian kernel function with bandwidth of 0.001 and $q = 100$ was used for discretization of the pdf. The authors of 3BE have extended their method for multiple input functions (see joint motion prediction experiment in Oliva et al. (2015)). We followed their method and concatenated the basis coefficients obtained from the three input distributions. In addition, for 3BE we scale the return samples to [0, 1] before applying cosine basis projection. The predicted distribution is then scaled back to the original range for quantification of the regression performance.

First, we performed evaluations for the task of predicting the next-day distributions. As we do not have the underlying true pdf for this real-world dataset, the regression performance is measured by the log-likelihood of the test samples. Table 2 shows the test log-likelihoods, where higher log-likelihood is favorable. Interestingly, the single-layer network in DRN (see Fig. 6) was sufficient to perform well, using just 7 network parameters. In comparison, MLP and 3BE require 4110 and 8100 parameters respectively.

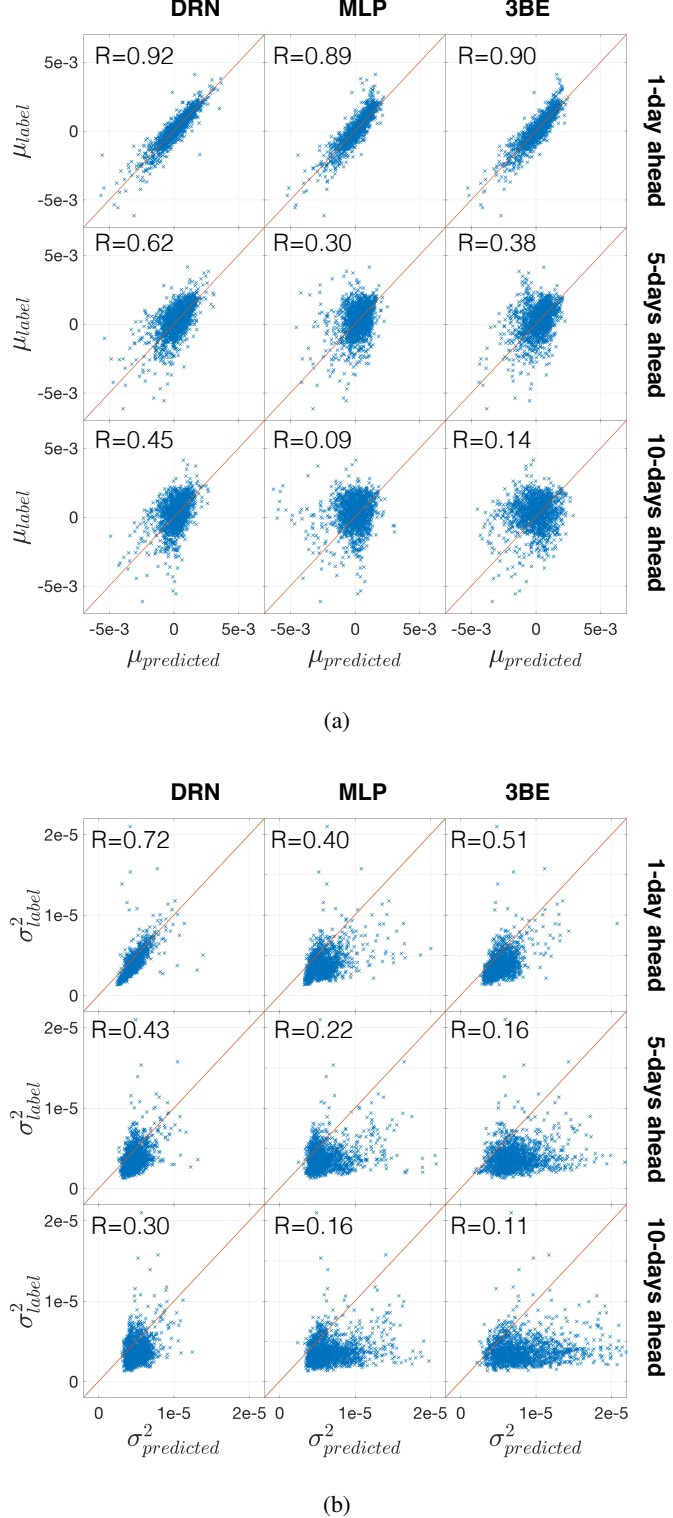

Figure 7: Comparison of the (a) mean and (b) variance of the label distributions and predicted distributions on the test set, for various $k$-days ahead predictions. The diagonal line represents a perfect fit where the predicted and labelled moments are equal. DRN outperforms the rest as its data points are closest to the diagonal line and it has the highest correlation coefficient (denoted by R) for all experiments.

To visualize the regression results on the test set, we compare for each day the first two moments (mean and variance) of the predicted distribution and the ground truth (see 1-day ahead panels of Fig. 7a and Fig. 7b). Each point represents one test data and we show the Pearson correlation coefficients between the predicted and labelled moments. DRN has the best regression performance as the points lie closest to the diagonal line where the predicted and labelled moments are equal, and its correlation values are highest.

### 3.3.1 PREDICTING SEVERAL DAYS AHEAD

As an extension, we predict the FTSE returns distribution several days ahead. The second and third rows of Fig. 7a and Fig. 7b show the moment plots for 5 and 10 days ahead respectively. Expectedly, the performance deteriorates as the number of days increases. Still, DRN outperforms the rest as shown by the moment plots and the correlation values. Fig. 8 summarizes the results by showing the average absolute error of the mean and variance as the number of days-ahead increases. For all experiments, DRN consistently has the lowest error.

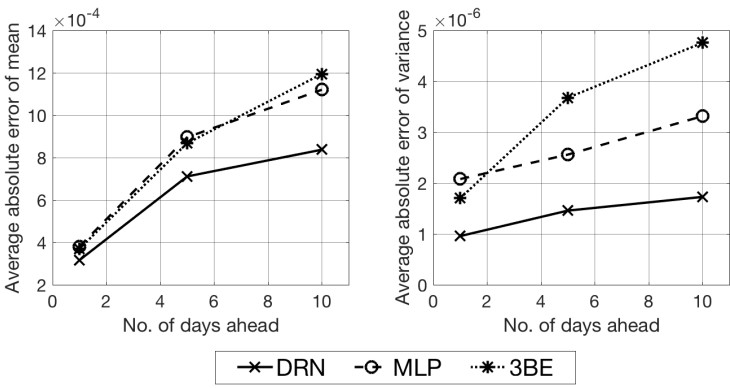

Figure 8: The average absolute error of mean and variance across the three methods for prediction with varying number of days-ahead. DRN's error is consistently the lowest compared to the benchmark methods. The standard errors are smaller than the data point symbols.

## 4 CELL LENGTH DATA

Finally, we conducted experiments on a real-world cell dataset similar to the one used in Oliva et al. (2013). The dataset is a time-series of images of NIH3T3 fibroblast cells. There are 277 time frames taken at 5-minute intervals, containing 176 to 222 cells each. In each frame, we measured the long and short-axis nuclear length of the cells and scaled the lengths to [0, 1]. At each time-frame, given the distribution of long-axis length, we predict the distribution of the short-axis length. The first 200 frames were used for training and last 77 for testing. For DRN and MLP, the pdf is estimated using kernel density estimation with a gaussian kernel of bandwidth 0.02 and $q = 100$ was used for discretization. We compare the log-likelihood on test data in Table 3. DRN had the best log-likelihood with a simple network of one input node connecting to one output node. In contrast, MLP and 3BE used more model parameters but achieved lower log-likelihoods. This validated DRN's advantage at learning distribution regressions on real-world data with fewer model parameters.

Table 3: Comparison of log-likelihood on the cell data and the number of model parameters.

|  | Log-likelihood on test set | Model description | No. of parameters |
|---|---|---|---|
| DRN | $148.50 \pm 0.46$ | No hidden layer | 5 |
| MLP | $147.80 \pm 0.10$ | 1 hidden layer of 20 nodes | 4120 |
| 3BE | $139.75 \pm 3.73$ | 9 projection coefficients, 5 Random Kitchen Sink features | 45 |

## 5 DISCUSSION

The distribution-to-distribution regression task has many useful applications ranging from population studies to stock market prediction. In this paper, we propose our Distribution Regression Network which generalizes the MLP framework by encoding a probability distribution in each node.

Our DRN is able to learn the regression mappings with fewer model parameters compared to MLP and 3BE. MLP has not been used for distribution-to-distribution regression in literature and we have adapted it for this task. Though both DRN and MLP are network-based methods, they encode the distribution very differently. By generalizing each node to encode a distribution, each distribution in DRN is treated as a single object which is then processed by the connecting weight. Thus, the propagation behavior in DRN is much richer, enabling DRN to represent the regression mappings with fewer parameters. In 3BE, the number of model parameters scales linearly with the number of projection coefficients of the distributions and number of Random Kitchen Sink features. In our experiments, DRN is able to achieve similar or better regression performance using less parameters than 3BE. Furthermore, the runtime for DRN is competitive with other methods (see comparison of mean prediction times in Appendix C).

For future work, we look to extend DRN for variants of the distribution regression task such as distribution-to-real regression and distribution classification. Extensions may also be made for regressing multivariate distributions.

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

# A   DERIVATIONS OF COST GRADIENTS

In Section 2.4, we presented the key equation for deriving backpropagation gradients:

$$\frac{\partial P_1^{(L)}\left(s_1^{(L)}\right)}{\partial P_k^{(l)}\left(s_k^{(l)}\right)} = \sum_j^n \sum_{s_j^{(l+1)}} \frac{\partial P_1^{(L)}\left(s_1^{(L)}\right)}{\partial P_j^{(l+1)}\left(s_j^{(l+1)}\right)} \frac{\partial P_j^{(l+1)}\left(s_j^{(l+1)}\right)}{\partial P_k^{(l)}\left(s_k^{(l)}\right)} \tag{10}$$

To derive the cost gradients for optimization, we need the derivative of the final output node distribution with respect to the network parameters. For instance, for the network weights:

$$\frac{\partial P_1^{(L)}\left(s_1^{(L)}\right)}{\partial w_{ki}^{(l)}} = \sum_{s_k^{(l)}} \frac{\partial P_1^{(L)}\left(s_1^{(L)}\right)}{\partial P_k^{(l)}\left(s_k^{(l)}\right)} \frac{\partial P_k^{(l)}\left(s_k^{(l)}\right)}{\partial w_{ki}^{(l)}} \tag{11}$$

We first derive the intermediate gradient terms required to obtain the gradients in Eq. (10) and proceed to derive gradients for the network parameters. All of the equations work on the discretized distributions, where the integrals are now expressed in summations.

## A.1   DERIVATION OF INTERMEDIATE BACKPROPAGATION TERMS

We start with deriving the final term of Eq. (10) which is the gradient of the upper layer node distribution with respect to the incoming lower node distribution. The subscripts and superscripts are renamed for ease of explanation in later derivations.

$$\frac{\partial P_k^{(l)}\left(s_k^{(l)}\right)}{\partial P_i^{(l-1)}\left(s_i^{(l-1)}\right)} = \sum_{s_k^{(l)'}} \frac{\partial P_k^{(l)}\left(s_k^{(l)}\right)}{\partial \tilde{P}_k^{(l)}\left(s_k^{(l)'}\right)} \frac{\partial \tilde{P}_k^{(l)}\left(s_k^{(l)'}\right)}{\partial P_i^{(l-1)}\left(s_i^{(l-1)}\right)} \tag{12}$$

The above derivative is a consequence of the normalization step taken in Eq. (7). At each node, we need to compute the derivative of the normalized distribution with respect to the unnormalized distribution. Recall $P_k^{(l)}\left(s_k^{(l)}\right) = \frac{\tilde{P}_k^{(l)}\left(s_k^{(l)}\right)}{Z_k^{(l)}}$, where $Z_k^{(l)} = \sum_{s_k^{(l)'}} \tilde{P}_k^{(l)}\left(s_k^{(l)'}\right)$.

$$\frac{\partial P_k^{(l)}\left(s_k^{(l)}\right)}{\partial \tilde{P}_k^{(l)}\left(s_k^{(l)'}\right)} = \begin{cases} \frac{1}{Z_k^{(l)}} - \frac{\tilde{P}_k^{(l)}\left(s_k^{(l)}\right)}{\left(Z_k^{(l)}\right)^2} & \text{if } s_k^{(l)} = s_k^{(l)'} \\ -\frac{\tilde{P}_k^{(l)}\left(s_k^{(l)}\right)}{\left(Z_k^{(l)}\right)^2} & \text{otherwise} \end{cases} \tag{13}$$

For the final term of Eq. (12), $\frac{\partial \tilde{P}_k^{(l)}\left(s_k^{(l)'}\right)}{\partial P_i^{(l-1)}\left(s_i^{(l-1)}\right)}$, the derivation proceeds from the propagation step in Eq. (5), reproduced here in discrete form.

$$\tilde{P}_k^{(l)}\left(s_k^{(l)}\right) = \exp\left(B\left(s_k^{(l)}\right)\right) \prod_i^n \left\{ \sum_{s_i^{(l-1)}} P_i^{(l-1)}\left(s_i^{(l-1)}\right) \exp\left[-w_{ki}^{(l)}\left(\frac{s_k^{(l)} - s_i^{(l-1)}}{\Delta}\right)^2\right] \right\} \tag{14}$$

By substituting

$$\beta\left(s_k^{(l)}\right) = \exp\left(B\left(s_k^{(l)}\right)\right),$$

$$\gamma_i\left(s_k^{(l)}\right) = \sum_{s_i^{(l-1)}} P_i^{(l-1)}\left(s_i^{(l-1)}\right) \exp\left[-w_{ki}^{(l)}\left(\frac{s_k^{(l)} - s_i^{(l-1)}}{\Delta}\right)^2\right],$$

we obtain

$$\tilde{P}_k^{(l)}\left(s_k^{(l)}\right) = \beta\left(s_k^{(l)}\right) \prod_i^n \gamma_i\left(s_k^{(l)}\right). \tag{15}$$

Eq. (15) is a product of variables and its derivative with respect to any variable is obtained by product rule.

$$\frac{\partial \tilde{P}_k^{(l)}\left(s_k^{(l)}\right)}{\partial x} = \tilde{P}_k^{(l)}\left(s_k^{(l)}\right)\left(\frac{\frac{\partial \beta\left(s_k^{(l)}\right)}{\partial x}}{\beta\left(s_k^{(l)}\right)} + \sum_i^n \frac{\frac{\partial \gamma_i\left(s_k^{(l)}\right)}{\partial x}}{\gamma_i\left(s_k^{(l)}\right)}\right), \tag{16}$$

where $x$ can be one of the lower layer node probabilities $P_i^{(l-1)}\left(s_i^{(l-1)}\right)$, or one of the network parameters $(w_{ki}^{(l)},\ b_{a,k}^{(l)},\ b_{q,k}^{(l)},\ \lambda_{a,k}^{(l)},\ \lambda_{q,k}^{(l)})$. Now we can derive the final term of Eq. (12).

$$\frac{\partial \tilde{P}_k^{(l)}\left(s_k^{(l)}\right)}{\partial P_i^{(l-1)}\left(s_i^{(l-1)}\right)} = \tilde{P}_k^{(l)}\left(s_k^{(l)}\right)\frac{\frac{\partial \gamma_i\left(s_k^{(l)}\right)}{\partial P_i^{(l-1)}\left(s_i^{(l-1)}\right)}}{\gamma_i\left(s_k^{(l)}\right)} \tag{17}$$

$$= \tilde{P}_k^{(l)}\left(s_k^{(l)}\right)\frac{\exp\left[-w_{ki}^{(l)}\left(\frac{s_k^{(l)}-s_i^{(l-1)}}{\Delta}\right)^2\right]}{\gamma_i\left(s_k^{(l)}\right)}$$

## A.2 Derivation of gradients with respect to network parameters

The derivatives of the unnormalized probability distribution of a node with respect to the connecting weights and bias parameters can be derived from Eq. (16).

First, for each node, we compute the derivative of its unnormalized distribution with respect to an incoming weight.

$$\frac{\partial \tilde{P}_k^{(l)}\left(s_k^{(l)}\right)}{\partial w_{ki}^{(l)}} = \frac{\tilde{P}_k^{(l)}}{\gamma_i\left(s_k^{(l)}\right)}\frac{\partial \gamma_i\left(s_k^{(l)}\right)}{\partial w_{ki}^{(l)}}, \tag{18}$$

where

$$\frac{\partial \gamma_i\left(s_k^{(l)}\right)}{\partial w_{ki}^{(l)}} = \sum_{s_i^{(l-1)}} P_i^{(l-1)}\left(s_i^{(l-1)}\right)\exp\left[-w_{ki}^{(l)}\left(\frac{s_k^{(l)}-s_i^{(l-1)}}{\Delta}\right)^2\right]\left[-\left(\frac{s_k^{(l)}-s_i^{(l-1)}}{\Delta}\right)^2\right]. \tag{19}$$

Similarly, for the bias parameters, we derive the gradients from Eq. (16). Here we show for $b_{a,k}^{(l)}$,

$$\frac{\partial \tilde{P}_k^{(l)}\left(s_k^{(l)}\right)}{\partial b_{a,k}^{(l)}} = \frac{\tilde{P}_k^{(l)}\left(s_k^{(l)}\right)}{\beta\left(s_k^{(l)}\right)}\frac{\partial \beta\left(s_k^{(l)}\right)}{\partial b_{a,k}^{(l)}}, \tag{20}$$

where

$$\frac{\partial \beta\left(s_k^{(l)}\right)}{\partial b_{a,k}^{(l)}} = \frac{\partial \exp\left(B\left(s_k^{(l)}\right)\right)}{\partial b_{a,k}^{(l)}} = \frac{\partial B\left(s_k^{(l)}\right)}{\partial b_{a,k}^{(l)}}\beta\left(s_k^{(l)}\right), \tag{21}$$

and

$$\frac{\partial B\left(s_k^{(l)}\right)}{\partial b_{a,k}^{(l)}} = \begin{cases} -\frac{s_k^{(l)}-\lambda_{a,k}^{(l)}}{\Delta} & \text{if } s_k^{(l)} > \lambda_{a,k}^{(l)} \\ \frac{s_k^{(l)}-\lambda_{a,k}^{(l)}}{\Delta} & \text{otherwise} \end{cases} \tag{22}$$

The derivatives for the other bias parameters can be obtained similarly.

$$\frac{\partial B\left(s_k^{(l)}\right)}{\partial b_{q,k}^{(l)}} = -\left(\frac{s_k^{(l)}-\lambda_{q,k}^{(l)}}{\Delta}\right)^2 \tag{23}$$

$$\frac{\partial B\left(s_k^{(l)}\right)}{\partial \lambda_{a,k}^{(l)}} = \begin{cases} b_{a,k}^{(l)} & \text{if } s_k^{(l)} > \lambda_{a,k}^{(l)} \\ -b_{a,k}^{(l)} & \text{otherwise} \end{cases} \tag{24}$$

$$\frac{\partial B\left(s_k^{(l)}\right)}{\partial \lambda_{q,k}^{(l)}} = -2b_{q,k}^{(l)} \left(\frac{s_k^{(l)} - \lambda_{q,k}^{(l)}}{\Delta}\right)\left(-\frac{1}{\Delta}\right) \tag{25}$$

$$= \frac{2b_{q,k}^{(l)}}{\Delta^2}\left(s_k^{(l)} - \lambda_{q,k}^{(l)}\right)$$

## B  DRN NETWORK ARCHITECTURE FOR SYNTHETIC DATASET

In this section, show the DRN network architecture used for the synthetic dataset results presented in Fig. 4a. There is one input node and one output node connected by a number of hidden layers of arbitrary width. All layers are fully-connected.

Table 4: DRN network architecture for the models presented in Fig. 4a. The network architecture is denoted as such: Eg. 1 - 4x3 - 1: 1 input node, followed by 4 layers each having 3 nodes, and 1 output node

| No. of model parameters | L2 test loss | DRN network architecture |
|---|---|---|
| 85 | 0.69 | 1 - 4x3 - 1 |
| 340 | 0.45 | 1 - 4x8 - 1 |
| 624 | 0.42 | 1 - 5x10 - 1 |
| 2044 | 0.37 | 1 - 5x20 - 1 |
| 3484 | 0.37 | 1 - 8x20 - 1 |

## C  COMPARISON OF PREDICTION TIMES

We compare the mean prediction time per data for DRN and the baseline methods. All runs were conducted on the CPU. For the synthetic dataset, we have shown the test loss for varying parameter sizes. For a fair comparison of runtime, for each method we chose a model size which gave a test L2 loss of about 0.37. For all the datasets, MLP has the fastest prediction time, followed by DRN and then 3BE.

Table 5: Comparison of mean prediction time per data for the experiments.

| | Mean prediction time per data / ms | | | |
|---|---|---|---|---|
| | Synthetic data | Ornstein-Uhlenbeck process | Stock data | Cell data |
| DRN | 1.65 | 0.64 | 0.59 | 0.29 |
| MLP | 0.03 | 0.01 | 0.01 | 0.06 |
| 3BE | 4.69 | 0.98 | 0.88 | 0.32 |

