# OpenReview forum: "Distribution Regression Network"
_ICLR.cc/2018/Conference — Reject_

### Official Review · AnonReviewer2 · 2017-11-27
**Potentially promising paper but difficult to see the practical significance**

**Rating:** 5
**Confidence:** 4

**Review:**

The paper considers distribution to distribution regression with MLPs.  The authors use an energy function based approach.  They test on a few problems, showing similar performance to other distribution to distribution alternatives, but requiring fewer parameters.

This seems to be a nice treatment of distribution to distribution regression with neural networks. The approach is methodological similar to using expected likelihood kernels.  While similar performance is achieved with fewer parameters, it would be more enlightening to consider accuracy vs runtime instead of accuracy vs parameters.  That’s what we really care about.  In a sense, because this problem has been considered several times in slightly different model classes, there really ought to be a pretty strong empirical investigation.  In the discussion, it says
“For future work, a possible study is to investigate what classes of problems DRN can solve.”  It feels like in the present work there should have been an investigation about what classes of problems the DRN can solve.  Its practical utility is questionable.  It’s not clear how much value there is adding yet another distribution to distribution regression approach, this time with neural networks, without some pretty strong motivation (which seems to be lacking), as well as experiments.  In the introduction, it would also improve the paper to outline clear points of methodological novelty.

---

> ### Author Response · Authors · 2017-12-15
> **Reply to Reviewer2**
>
> We appreciate your feedback and suggestions. We address your concerns in the following points:
>
> 1) While similar performance is achieved with fewer parameters, it would be more enlightening to consider accuracy vs runtime instead of accuracy vs parameters.
>
> Thank you for the suggestion, we have included a comparison of the runtimes in Appendix C. DRN’s runtime is competitive compared to the other methods. MLP has the fastest prediction time, followed by DRN and then 3BE.
>
> 2) It feels like in the present work there should have been an investigation about what classes of problems the DRN can solve.
>
> In this paper, we have shown DRN to work well for various forms of univariate distributions (eg. unimodal and bimodal distributions, symmetric and assymetric distributions), and for a variety of mappings as described in the synthetic dataset (peak splitting, peak shifting, peak spreading, peak coalescence). For future work, we look to extend DRN other classes of the distribution regression task (eg. distribution-to-real, distribution classification) and to handle multivariate distributions. We have improved on our explanation of future work in the conclusion.
>
> 3) It’s not clear how much value there is adding yet another distribution to distribution regression approach, this time with neural networks, without some pretty strong motivation (which seems to be lacking), as well as experiments.
>
> We appreciate your concern on the value of our proposed method in comparison with the existing works. We address them in the following, and have included the comparisons in the revised manuscript. As far as we know, these are the works related to distribution-to-distribution regression which we have cited in our paper:
> 1.	Linear Smoother Estimator (LSE) by Olivia et. al., 2013
> 2.	Triple-Basis Estimator (3BE) by Olivia et. al., 2015
> 3.	Extrapolating the Distribution Dynamics (EDD) by Lampert 2015
> 4.	Multilayer perceptron (MLP): has not been used for distribution-to-distribution regression in literature, we have adapted it for this task
>
> LSE is a instance-based learning method which does not scale well with data size, whereas the inference time of our proposed method is independent of training size. EDD addresses a specific task of predicting the future state of a time-varying probability distribution and it is unclear how it can be used for a more general case of regressing between distributions of different objects.
>
> In addition, LSE and EDD are designed for single input, single output regressions and their effectiveness on multiple input, multiple output distributions is unclear. In comparison, our proposed Distribution Regression Network’s (DRN) network architecture gives it the flexibility to handle arbitrary number of input and output distributions.
>
> Distribution Regression Network is able to learn the regression mappings with fewer model parameters compared to 3BE and MLP. In 3BE, the number of model parameters scales linearly with the number of projection coefficients of the distributions and the number of Random Kitchen Sink features. From our experiments, DRN is able to achieve similar or better regression performance using less parameters than 3BE.
>
> Though both DRN and MLP are network-based methods, they encode the distribution very differently – in DRN, each node encodes a distribution while in MLP, each node encodes a real number corresponding to a bin of the distribution. By generalizing each node to encode a distribution, each distribution is treated as a single object which is then processed by the connecting weight. Thus, the propagation behavior in DRN is much richer, enabling DRN to represent the regression mappings with fewer parameters.
>
> 4) In the introduction, it would also improve the paper to outline clear points of methodological novelty.
>
> Thank you for the suggestion. Our methodological novelty is as discussed in the earlier comparison with other methods. We have added points of methodological novelty in the introduction.

---

### Official Review · AnonReviewer1 · 2017-11-29
**This paper introduces a simple but neat network architecture for representing probability distributions that is supported by preliminary experiments. However, the paper would be much stronger with more robust evaluation on real-world data.**

**Rating:** 7
**Confidence:** 2

**Review:**

Summary:

This paper presents a new network architecture for learning a regression of probability distributions.

The distribution output from a given node is defined in terms of a learned conditional probability function, and the output distributions of its input nodes. The conditional probability function is an unnormalized distribution with the same form as the Boltzman distribution, and distributions are approximated from point estimates by discretizing the finite support into predefined equal-sized bins. By letting the conditional distribution between nodes be unnormalized, and using an energy function that incorporates child nodes independently, the approach admits efficient computation that does not need to model the interaction between the distributions output by nodes at a given level.

Under these dynamics and discretization, the chain rule can be used to derive a matrix of gradients at each node that denotes the derivative of the discretized output distribution with respect to the current node's discretized distribution. These gradients are in turn used to calculate updates for the network parameters with respect to the Jensen Shannon divergence between the predicted distribution and a target distribution.

The approach is evaluated on three tasks, two synthetic and one real world. The baselines are the state of the art triple basis estimator (3BE) or a standard MLP that represents the output distribution using a softmax over quantiles. On both of the synthetic tasks --- which involve predicting gaussians --- the proposed approach can fit the data reasonably using far fewer parameters than the baselines, although 3BE does achieve better overall performance. On a real world task that involves predicting a distribution of future stock market prices from multiple input stock marked distributions, the proposed approach significantly outperforms both baselines. However, this experiment uses 3BE outside of its intended use case --- which is for a single input distribution --- so it's not entirely clear how well the very simple proposed model is doing.

Notes to authors:

I'm not familiar with 3BE but the fact that it is used outside of its intended use case for the stock data is worrying. How does 3BE perform at predicting the FTSE distribution at time t + k from the FTSE distribution at time t only? Do the multiple input distributions actually help?

You use a kernel density estimate with a Gaussian kernel function to estimate the stock market pdf, but then you apply your network directly to this estimate. What would happen if you built more complex networks using the kernel values themselves as inputs?

Could you also run experiments on the real-world datasets used by the 3BE paper?

What is the structure of the DRN that uses > 10^3 parameters (from Fig. 4)? The width of the network is bounded by the two input distributions, so is this network just incredibly deep? Also, is it reasonable to assume that both the DRN and MLP are overfitting the toy task when they have access to an order of magnitude more parameters than datapoints.

It would be nice if section 2.4 was expanded to actually define the cost gradients for the network parameters, either in line or in an appendix.

---

> ### Author Response · Authors · 2017-12-15
> **Reply to Reviewer1**
>
> Thank you for your comments and suggestions. Here are our replies:
>
> 1) However, this experiment uses 3BE outside of its intended use case --- which is for a single input distribution --- so it's not entirely clear how well the very simple proposed model is doing.
>
> We would like to clarify that the authors of 3BE performed a regression on multiple input functions for one of their experiments (see joint motion prediction experiment in Olivia et. al, 2015).  We followed their method to extend to multiple input distributions by concatenating the basis coefficients from the input distributions. Hence, this is not considered using 3BE outside of its intended use case. We have updated our manuscript to explain this more clearly, in Section 3.3, paragraph 4.
>
> 2) For the stock dataset, we reported that 3BE’s predicted distributions have means that are near zero for all test data. We found out this was because we followed the 3BE paper and mistakenly applied cosine basis estimator directly on the return samples without any pre-processing. The stock return values are small (range of [-0.015, 0.015]) and centered around zero, and cosine basis functions are symmetric with respect to the y-axis. Hence, all of the estimated distributions are centred about zero. We have corrected this by rescaling the returns to range of [0, 1] before applying cosine basis projection. The new results are shown in our revision. While 3BE’s accuracy has improved and the predicted distribution means are more reasonable (see Fig. 7), overall 3BE’s accuracy still lags behind DRN and MLP.
>
> 3) How does 3BE perform at predicting the FTSE distribution at time t + k from the FTSE distribution at time t only? Do the multiple input distributions actually help?
>
> For single input distribution, 3BE’s performance is still lagging behind the other methods. We observe that having multiple inputs improved the accuracy for DRN, but not for the other two methods.
>
> Log-likelihood on test set for next-day prediction:
> DRN single input:     474.37 +- 0.01
> DRN multiple input: 474.43 +- 0.01
> MLP single input:     471.81 +- 0.09
> MLP multiple input: 471.50 +- 0.08
> 3BE single input:      467.46 +- 0.93
> 3BE multiple input:  466.76 +- 0.73
>
> 4) What would happen if you built more complex networks using the kernel values themselves as inputs?
>
> We would like to clarify what you meant by ‘kernel values’ in your comments. Do you mean to feed the companies returns directly into a more complex network without first estimating the distribution?
>
> 5) Could you also run experiments on the real-world datasets used by the 3BE paper?
>
> The 3BE paper experiments on multivariate distributions. However, our current model is designed for univariate distributions; multivariate distributions are planned for future work. The LSE paper (Oliva et. al., 2013) experimented on a dataset of time series microscopy images of cells. Each image frame contains a number of cells. For each time frame, given the distribution of the long-axis length of cells, they predict the distribution of short-axis length. We are currently running experiments on a similar dataset and will add it to our paper.
>
> 6) What is the structure of the DRN that uses > 10^3 parameters (from Fig. 4)? The width of the network is bounded by the two input distributions, so is this network just incredibly deep?
>
> We would like to clarify that for the first synthetic dataset, each input and output distribution is a weighted sum of two Gaussians and not simply a Gaussian distribution, as illustrated in Fig. 3(b). Hence there is one input node and one output node connected by a hidden layers of arbitrary width. All layers are fully-connected. We have included the network structure for DRN in Appendix B.
>
> 7) Also, is it reasonable to assume that both the DRN and MLP are overfitting the toy task when they have access to an order of magnitude more parameters than datapoints.
>
> For DRN and MLP, there is no significant overfitting as the gaps between train and test loss are not significant. We have included the training losses in Fig. 4(b).
>
> 8) It would be nice if section 2.4 was expanded to actually define the cost gradients for the network parameters, either in line or in an appendix.
>
> Thank you for the suggestion, we have expanded the cost gradient derivations and included them in Appendix A.

---

> > ### Author Response · Authors · 2017-12-20
> > **Additional real-world dataset included**
> >
> > We have completed our experiment on the real-world cell dataset used by the LSE paper (Oliva et. al., 2013) and have added it to Section 4 of our paper. Similar to the findings in the stock dataset, DRN achieved the highest log-likelihood compared to MLP and 3BE, using fewer model parameters.

---

### Official Review · AnonReviewer3 · 2017-11-30
**Distribution Regression Network**

**Rating:** 7
**Confidence:** 4

**Review:**

This is an intriguing paper on running regressions on probability distributions: i.e. a target distribution is expressed as a function of input distributions. A well-written manuscript, though the introduction could have motivated the problem a little better (i.e. why would we want to do this). The novelty in the paper is implementing such a regression in a layered network. The paper shows how the densities at each nodes are computed (and normalised). Optimisation by back propagation and discretization of the densities to carry out numerical integration are well explained and easy to follow. The paper uses three problems to illustrate the idea -- a synthetic dataset, a mean reverting stochastic process and a prediction problem on stock indices.
My only two reservations of this paper is the illustration on the stock index data -- it seems to me, returns on individual constituent stocks of an index are used as samples of the return on the index itself.  But this cannot be true when the index is a weighted sum of the constituent assets.  Secondly, it is not clear to me why one would force a kernel density estimate on the asset returns and then bin the density into 100 bins for numerical reasons -- does the smoothing that results from this give any advantage over a histogram of the returns in 100 bins?

---

> ### Author Response · Authors · 2017-12-15
> **Reply to Reviewer3**
>
> We appreciate your succinct summary of our paper and your comments. We have added in stronger motivations for the distribution-to-distribution task and better explanation of our model’s novelty. Please refer to the introduction of the revised paper.
>
> 1) it seems to me, returns on individual constituent stocks of an index are used as samples of the return on the index itself. But this cannot be true when the index is a weighted sum of the constituent assets.
>
> The stock index is a single number defined by the weighted sum of constituent stock prices. In this paper, we are not concerned about the return of the stock index, nor to form a distribution of it.  Instead, we work on the distribution of the stock returns of constituent companies of the index, with individual company returns used as samples. We have provided a more accurate description of the dataset in the revised paper, in Section 3.3.
>
> 2) it is not clear to me why one would force a kernel density estimate on the asset returns and then bin the density into 100 bins for numerical reasons -- does the smoothing that results from this give any advantage over a histogram of the returns in 100 bins?
>
> We use kernel density estimate on the return samples as it provides a principled way to account for the uncertainty in sample measurements, where the kernel width correlates to the extent of uncertainty. In the case of measuring stock returns, we use the closing price to estimate the daily stock price of a company. Furthermore, a practical reason is that using histogram binning may result in empty bins especially when number of samples is small. It also causes discontinuities in the estimation of the probability distribution.

---

### Author Response · Authors · 2017-12-15
**Response to all reviewers**

Thank you for your constructive comments. We have replied to the individual reviews separately and revised our paper accordingly.

We would like to highlight that we have previously misrepresented 3BE’s performance on the stock dataset. We reported that 3BE’s predicted distributions have means that are near zero for all test data. We found out this was because we followed the 3BE paper and mistakenly applied cosine basis estimator directly on the return samples without any pre-processing. The stock return values are small (range of [-0.015, 0.015]) and centered around zero, and cosine basis functions are symmetric with respect to the y-axis. Hence, all of the estimated distributions are centred about zero. We have corrected this by rescaling the returns to range of [0, 1] before applying cosine basis projection. The new results are shown in our revision. While 3BE’s accuracy has improved and the predicted distribution means are more reasonable (see Fig. 7), overall 3BE’s accuracy still lags behind DRN and MLP.

We have also conducted more runs over different random seeds for all three methods on the stock dataset to obtain lower standard errors.

---

### Decision · Program_Chairs · 2018-01-29
**ICLR 2018 Conference Acceptance Decision**

**Decision:**

Reject

**Comment:**

The paper proposes a method to map input probability distributions to output probability distributions with few parameters. They show the efficacy of their method on synthetic and real stock data. After revision they seemed to have added another dataset, however, it is not carefully analyzed like the stock data. More rigorous experimentation needs to be done to justify the method.